# Voriconazole Eye Drops: Enhanced Solubility and Stability through Ternary Voriconazole/Sulfobutyl Ether β-Cyclodextrin/Polyvinyl Alcohol Complexes

**DOI:** 10.3390/ijms24032343

**Published:** 2023-01-25

**Authors:** Hay Man Saung Hnin Soe, Khanittha Kerdpol, Thanyada Rungrotmongkol, Patamaporn Pruksakorn, Rinrapas Autthateinchai, Sirawit Wet-osot, Thorsteinn Loftsson, Phatsawee Jansook

**Affiliations:** 1Faculty of Pharmaceutical Sciences, Chulalongkorn University, 254 Phyathai Road, Pathumwan, Bangkok 10330, Thailand; 2Center of Excellence in Biocatalyst and Sustainable Biotechnology, Department of Biochemistry, Faculty of Science, Chulalongkorn University, 254 Phyathai Road, Pathumwan, Bangkok 10330, Thailand; 3Program in Bioinformatics and Computational Biology, Graduate School, Chulalongkorn University, Bangkok 10330, Thailand; 4Department of Medical Sciences, Ministry of Public Health, Tiwanon Road, Nonthaburi 11000, Thailand; 5Faculty of Pharmaceutical Sciences, University of Iceland, Hofsvallagata 53, IS-107 Reykjavik, Iceland

**Keywords:** cyclodextrin, voriconazole, eye drops, stability, solubility, ternary complex

## Abstract

Voriconazole (VCZ) is a broad-spectrum antifungal agent used to treat ocular fungal keratitis. However, VCZ has low aqueous solubility and chemical instability in aqueous solutions. This study aimed to develop VCZ eye drop formulations using cyclodextrin (CD) and water-soluble polymers, forming CD complex aggregates to improve the aqueous solubility and chemical stability of VCZ. The VCZ solubility was greatly enhanced using sulfobutyl ether β-cyclodextrin (SBEβCD). The addition of polyvinyl alcohol (PVA) showed a synergistic effect on VCZ/SBEβCD solubilization and a stabilization effect on the VCZ/SBEβCD complex. The formation of binary VCZ/SBEβCD and ternary VCZ/SBEβCD/PVA complexes was confirmed by spectroscopic techniques and in silico studies. The 0.5% *w*/*v* VCZ eye drop formulations were developed consisting of 6% *w*/*v* SBEβCD and different types and concentrations of PVA. The VCZ/SBEβCD systems containing high-molecular-weight PVA prepared under freeze–thaw conditions (PVA-H hydrogel) provided high mucoadhesion, sustained release, good ex vivo permeability through the porcine cornea and no sign of irritation. Additionally, PVA-H hydrogel was effective against the filamentous fungi tested. The stability study revealed that our VCZ eye drops provide a shelf-life of more than 2.5 years at room temperature, while a shelf-life of only 3.5 months was observed for the extemporaneous Vfend^®^ eye drops.

## 1. Introduction

Fungal keratitis is one of the leading causes of blindness in developing countries [1]. The infections are associated with ocular trauma caused by vegetative matter or soil-contaminated objects. In recent years, the rise in contact lens wearing, especially in developed nations, has also contributed to a risk factor for fungal keratitis [2]. Filamentous fungi, such as *Fusarium* and *Aspergillus*, are the most common pathogenic microorganisms of mycotic keratitis. The treatment of fungal keratitis is complicated and challenging due to delayed diagnosis and likely unpredictable susceptibility of fungi to antifungal drugs [3]. Moreover, only one commercial topical antifungal drug (natamycin eye drops) is available. The prescription of additional topical antifungal preparations requires a compounding pharmacy, increasing the overall treatment cost [4]. Because fungal keratitis is likely to affect over one million people each year [5], a simple and precise diagnostic method plus the availability of affordable antifungal eye drops are needed.

Voriconazole (VCZ) is a second-generation triazole antifungal agent. It disturbs the growth of fungi by inhibiting the cytochrome P450-dependent 14α-lanosterol demethylation step, which is important for the ergosterol synthesis of fungi [6,7]. Topical VCZ eye drops have been extemporaneously prepared and prescribed as off-label medication to treat keratitis, with promising results [2,8]. Lau et al. (2008) [9] studied the chemical stability of a 1% (*w*/*v*) VCZ eye drop solution, prepared by reconstituting Vfend^®^ powder with water for injection (WFI). When stored at 2 to 8 °C, the prepared VCZ eye drops were chemically stable for up to 75 days, and the predicted shelf-life was approximately 4 months. The resulting data were similar to the investigations by Al-Badriyeh and co-workers [10]. Table 1 displays the chemical structure and some physicochemical properties of VCZ. VCZ has low aqueous solubility (0.5 mg/mL at room temperature) and is unstable in aqueous solution, i.e., degradation by hydrolysis [11].

The use of cyclodextrin (CD) for inclusion complex formation is a promising technique to increase the aqueous solubility of poorly water-soluble drugs and enhance the chemical stability of labile compounds [12]. CDs are cyclic oligosaccharides possessing toroid structures with a hydrophilic outer surface and somewhat lipophilic central cavity. The common parent CDs, i.e., α-cyclodextrin (αCD), β-cyclodextrin (βCD) and γ-cyclodextrin (γCD), contain six, seven and eight glucopyranose units, respectively. CDs can increase the solubility of lipophilic drugs in aqueous solutions by inserting lipophilic moieties of drugs in their central cavities [13,14]. However, due to the limited aqueous solubility and complexing ability of natural CDs, the CD derivatives have been used for complexation with various water-insoluble drugs. For example, βCD derivatives, such as sulfobutyl ether β-cyclodextrin (SBEβCD) and 2-hydroxypropyl β-cyclodextrin (HPβCD), are better pharmaceutical solubilizers than the parent βCD, and more applicable to use in aqueous drug products [15]. 

In aqueous solutions, water-soluble polymers are able to improve the solubility of drug/CD complexes and enhance the chemical stability of drugs by changing the hydration property of CD or by forming complex aggregates [15,16,17]. Polyvinyl alcohol (PVA) is a biocompatible polymer that is commonly used in ophthalmic formulations [18,19]. When PVA was used as a ternary component, it protected the thermal degradation of eugenol/CD complexes [20]. The presence of chitosan (CS), a positively charged polysaccharide as a ternary compound, enhanced the solubility of progesterone/HPβCD complexes [21]. Our previous study also reported that CS increased the complexation of asiaticoside/SBEβCD by ternary complex nanoaggregate formation [22]. Hyaluronic acid (HA) is an anionic glycosaminoglycan of natural origin and is known as a biocompatible and biodegradable water-soluble polymer [23]. The synergistic effect of HA on CD solubilization of celecoxib, by forming a ternary celecoxib/CD/HA complex, was also reported [24]. Recently, Suvarna et al. (2021) [25] studied the inclusion interaction of VCZ with SBEβCD in the presence of polymers and introduced it as a promising strategy for enhancing the solubility of VCZ. However, to the best of our knowledge, none of the enhanced chemical stability of VCZ in the ternary complex has been reported elsewhere.

Thus, the aim of this study was to evaluate the aqueous solubility and chemical stability enhancement of VCZ through the formation of binary (VCZ/CD) and ternary (VCZ/CD/polymer) complexes. The CD inclusion complexes were characterized and the kinetic stability of VCZ in pure water and the complexing solutions was evaluated. Topical VCZ eye drop formulations were developed and the physicochemical properties, mucoadhesion, drug release and ex vivo permeation were investigated. Additionally, the cytocompatibility, antifungal activity and chemical stability of VCZ in eye drop formulations were assessed. 

## 2. Results and Discussion

### 2.1. Solubility Determinations

Figure 1 displays the phase-solubility profiles of VCZ in aqueous solutions containing various CDs. The VCZ concentration increased linearly as a function of αCD and βCD concentrations (Figure 1a,b). Thus, A_L_-type phase-solubility profiles were observed, indicating the formation of a 1:1 VCZ/CD complex [26]. In the case of γCD, a B_S_-type phase-solubility profile was observed (Figure 1c) that referred to the limited solubility of VCZ/γCD complexes at high aqueous γCD concentrations [27].

The values of the apparent stability constant (K_1:1_) and the complexation efficiency (CE) of VCZ/CD complexes are displayed in Table 2. VCZ had a 10-fold higher affinity to βCD than αCD. However, the molecular rigidity and strong intermolecular hydrogen bonding in the crystal state resulted in limited solubility of native βCD in an aqueous medium. Attachment of substituent groups can favor βCD interaction with the surrounding water molecules and the random distribution of the substituents, both of which will increase aqueous βCD solubility [28,29]. Hence, βCD derivatives, i.e., HPβCD, randomly methylated β-cyclodextrin (RMβCD), SBEβCD and carboxymethyl β-cyclodextrin (CMβCD), were further investigated for VCZ solubility enhancement. The phase-solubility diagrams of VCZ in all tested βCD derivatives represented the A_L_ type, which remained unchanged from its parent βCD (Figure 1d). The βCD derivative ranking regarding K_1:1_ and CE values was as follows: SBEβCD > HPβCD > RMβCD >> CMβCD (Table 2). The significant distance between SBEβCD torus and the charged sulfonate moiety reduced the steric interference and increased the binding potential when compared with the other βCD derivatives. In addition, the long alkyl chain length acted as a supplementary binding site for substrates [30,31]. Based on the solubility determination data, SBEβCD was selected for further studies.

The addition of water-soluble polymers to drug/CD complexes forms a ternary system via hydrophobic interaction, van der Waals dispersion forces or hydrogen bonds, thereby enhancing the CD solubilization of drugs [32,33]. In this study, PVA, CS and HA, classified as non-ionic, cationic and anionic polymers, respectively, were used. As expected, the addition of polymers forming ternary complexes increased the solubilizing capacity of the binary VCZ/SBEβCD complex (Table 2). Of these tested polymers, PVA exhibited the highest CE ratio, enhancing the complexation efficacy (CE) of SBEβCD by 30%. PVA, the water-soluble polymer, interacted with the outside surfaces of CD, forming aggregates or co-complexes, enhancing the poorly soluble drugs through micellar-type solubilization [34]. In the case of CS, the resulting CE increment might have been due to the additional electrostatic interaction between the protonated ammonium group of CS and the negatively charged sulfobutyl moiety of SBEβCD [22].

### 2.2. Particle Size, Size Distribution and Zeta Potential of Binary VCZ/SBEβCD and Ternary VCZ/SBEβCD/Polymer Complexes 

The particle size, size distribution and zeta potential of VCZ saturated in aqueous 10% w/v SBEβCD solutions in the presence and absence of individual polymers are displayed in Table 3. The aggregate sizes and size distributions of the complexes based on their respective intensity peaks are displayed in Appendix A. Three different size populations, i.e., small, medium and large aggregates, were assigned in both binary and ternary complex systems. The first population was smaller-size aggregates, which were 1–2 nm in diameter, represented to the binary VCZ/SBEβCD inclusion complexes. The second and third populations referred to the formation of binary VCZ/SBEβCD or ternary VCZ/SBEβCD/polymer complex aggregates, which varied from 10–1000 nm in diameter to a few micrometers [35]. Water-soluble polymers are well known to be able to enhance the solubilizing effect of CD complexes and stabilize self-assembled CD aggregates by forming CD–polymer hydrogen bonds [34]. 

Compared with the binary VCZ/SBEβCD complexes, adding either CS or HA exhibited larger aggregate sizes. Surprisingly, the presence of PVA in the complexing media showed the overall mean particle size to be smaller than that of the respective binary VCZ/SBEβCD complexes. PVA is an amphiphilic polymer that acts as a surface-active agent by lowering interfacial tensions, thus, decreasing aggregate sizes [36,37]. However, the medium and large aggregates were predominantly observed, corresponding to the ability of the aggregates to increase VCZ solubility.

The zeta potential of ternary systems containing CS or HA presented positive and negative values, respectively, which were associated with their characteristics in nature. The addition of PVA provided a significantly higher zeta potential value than that of the binary VCZ/SBEβCD complex. This may imply that the negatively charged PVA chains surrounding the CD complex lead to increased complex stabilization. Among the polymers tested, PVA provided the highest SBEβCD solubilization increment and proper nanoaggregate size and, thus, was selected for further studies.

### 2.3. Transmission Electron Microscopy (TEM) Analysis

The TEM image of the binary VCZ/SBEβCD complex displays particles with predominant particle sizes ranging from 1 to 2 nm, referred to as the inclusion complexes, as well as small VCZ/SBEβCD complex aggregates with particle sizes ranging from 10 to 20 nm (Figure 2a). In the case of the ternary VCZ/SBEβCD/PVA complex, larger nanoaggregates were observed (Figure 2b). It could be assumed that VCZ/SBEβCD complexes are localized as a dense core surrounded by the polymeric network of PVA chains. The particles obtained from TEM images were similar to the dynamic light scattering (DLS) results. This supported that PVA formed ternary complexes and stabilized their small complex aggregates via electrostatic and steric hindrance effects [35,38]. 

### 2.4. Fourier-Transform Infrared (FT-IR) Spectroscopy

FT-IR spectroscopy was used to determine the intermolecular interaction between the drug and CD upon complexation. The FT-IR spectrum of VCZ exhibited the principle absorption bands of C-N, C-F and C-C stretching at 3202.70, 1495.05 and 1585.12 cm^−1^, respectively (Figure 3a) [39]. SBEβCD frequencies were characterized by intense bands due to O-H stretching vibration at 3376.18 cm^−1^, C-H stretching at 2928.13 cm^−1^, C=O stretching at 1647.06 cm^−1^, C-O-C stretching vibration at 1150.40 cm^−1^ and a sharp band at 1032.69 cm^−1^, corresponding to sulfoxide stretch (Figure 3b) [40]. Simple overlapping of absorption bands was found in physical mixture (PM) of VCZ/SBEβCD (Figure 3c), while the characteristic absorption bands of VCZ disappeared in freeze-dried samples of the binary complex (Figure 3d). Additionally, the FT-IR bands of the O-H, C-H, C=O, C-O-C and S=O groups were shifted to 3354.07, 2918.63, 1645.94, 1154.42 and 1031.73, respectively. These observations were considered to be evidence for the presence of inclusion complexes, similar to a related report [25].

For the ternary VCZ/SBEβCD/PVA complex, the O-H stretching was significantly shifted to a lower frequency, i.e., wave number 3347.81 cm^−1^, while other characteristic peaks slightly changed compared with the binary VCZ/SBEβCD complex (Figure 3e). The spectra of binary VCZ/SBEβCD and ternary VCZ/SBEβCD/PVA complexes did not show new peaks, indicating that no chemical bonds were formed in these complexes. These spectral changes were possible due to the dissociation of the intermolecular hydrogen bonds of VCZ through the inclusion complex formation through the interactions between the functional groups of VCZ, SBEβCD and PVA [22,25,41].

### 2.5. Proton Nuclear Magnetic Resonance (^1^H-NMR) Spectroscopy

^1^H-NMR spectroscopy was carried out to identify the inclusion mode of VCZ included in the SBEβCD cavity [42]. The changes in proton chemical shift values (Δδ*) in binary VCZ/SBEβCD and ternary VCZ/SBEβCD/PVA complexes are summarized in Table 4.

In the presence of SBEβCD, the protons in the difluoro phenyl ring (H-1, H-2, H-3) and fluoropyrimidine ring (H-4, H-5) of VCZ in the binary complex showed significant upfield shifts. The signals corresponded to SBEβCD exhibiting the downfield shifts of inner cavity protons, i.e., H-3 (+0.002) and H-5 (+0.004). The chemical shifts of H-3 and H-5 of CD are crucial for possible complex formations between drug and CD [43]. Thus, it could be concluded that both the difluoro phenyl and fluoropyrimidine rings of VCZ were inserted in the hydrophobic SBEβCD cavity. 

The effect of PVA on orientational changes of VCZ/SBEβCD was further explored by observing the changes in their Δδ* values in the ternary VCZ/SBEβCD/PVA complex. Compared with the binary complex, additional upfield shifts of H-2 and H-4 (the outer-cavity protons of SBEβCD) were noted, possibly due to the hydrogen bonding interaction between SBEβCD and PVA. In addition, the H-3 and H-5 protons of SBEβCD exhibited a significant downfield shift with Δδ* values of +0.009 and +0.022, respectively, suggesting that the VCZ molecule was deeply included in the SBEβCD cavity. Interestingly, adding PVA significantly enhanced the upfield shift at H-6 of VCZ (−0.011). It has been reported that H-6 was located in the labile part of VCZ, where the degradation occurs via the cleavage of VCZ [11]. Therefore, it could be assumed that PVA could enhance VCZ stability through the shielding of its sensitive functional group in the ternary VCZ/SBEβCD/PVA complex formation.

### 2.6. Molecular Dynamics (MD) Simulations 

MD simulations and binding free energy calculations were performed to investigate the preferential binding mode of VCZ/SBEβCD and VCZ/SBEβCD/PVA inclusion complexes and to study the effect of PVA. To evaluate the stability of the inclusion complexes, RMSD of VCZ (gray), SBEβCD (red) and PVA (blue) is plotted in Figure 4a. We found that VCZ and SBEβCD of Form I (difluoro phenyl insertion) and Form II (fluoropyrimidine insertion), with and without PVA, showed similar trends in system stability with RMSD of ~1 Å and ~4–5 Å, respectively, while the RMSD of PVA was much higher than that of VCZ and SBEβCD throughout the simulations. Moreover, the R_gyr_ of the VCZ/SBEβCD and VCZ/SBEβCD/PVA of all systems was determined to roughly investigate the inclusion complex. The R_gyr_ values of the VCZ/SBEβCD (black) and VCZ/SBEβCD/PVA (green) of all forms were approximately 7 to 8 Å and 10 to 26 Å, respectively. These results indicated that VCZ was stable inside the SBEβCD’s hydrophobic inner cavity of all forms, while PVA was attached to the hydrophilic outer surface of SBEβCD throughout the simulations. In addition, to further investigate the solvent effect of PVA on the inclusion complexes, the water accessibility towards the VCZ molecule inside SBEβCD’s cavity over the last 20 nanoseconds (ns) simulation time was determined. The average SASA of Form I (~90 Å^2^) was lower than that of Form II (~108 Å^2^), suggesting that Form I was the preferred binding mode rather than Form II. Furthermore, the average SASA of the VCZ/SBEβCD/PVA in Forms I and II showed an average SASA of ~80 Å^2^ and 89 Å^2^, respectively, lower than those of the VCZ/SBEβCD in both forms. This indicated that PVA could be used as a stabilizer for VCZ, in which PVA hinders water from interacting with VCZ. This agreed with the number of atom contacts between VCZ and the host molecules because the maximum values of the number of atoms contacting both forms in the VCZ/SBEβCD/PVA (~15 to 289) was ~3-fold higher than those in the VCZ/SBEβCD (~15 to 87). 

It was illustrated that PVA could possibly occupy the VCZ/SBEβCD inclusion complex. The average MM/GBSA (*ΔG_MM/GBSA_*) and MM/PBSA (*ΔG_MM/PBSA_*) binding free energies of the inclusion complexes over the last 20 ns simulation times were calculated, and the represented snapshots are depicted in Figure 4b. It could be noted that the *ΔG_MM/GBSA_* and *ΔG_MM/PBSA_* of complexes exhibited mostly similar trends, and the *ΔG_MM/PBSA_* was chosen for discussion. Overall, the encapsulation processes of all systems were mainly driven by van der Waals interaction (−38.85 ± 0.18 to −42.38 ± 0.16 kcal/mol). In the case of Form I, the center of the VCZ molecule was tightly located inside the hydrophobic cavity of SBEβCD with a lower *ΔG_MM/PBSA_* of −8.26 ± 0.18 kcal/mol, whereas in Form II, VCZ remained near the primary rim of SBEβCD with a higher *ΔG_MM/PBSA_* of −4.10 ± 0.17 kcal/mol (Figure 4b). Furthermore, the binding affinity of the VCZ/SBEβCD/PVA was higher than that of the VCZ/SBEβCD, suggesting that PVA assisted the stability of the inclusion complex of VCZ/SBEβCD, especially Form II, correlating to the results of SASA and the number of atom contacts. It could be stated that these obtained results could support the experimental ^1^H-NMR data. 

### 2.7. Kinetic Degradation Studies of VCZ 

It has been reported that VCZ was degraded in aqueous solution under stress conditions, i.e., alkaline medium and elevated temperature [11]. Our results from ^1^H-NMR and computational data suggested that the liable part of VCZ was included inside the cavity of SBEβCD. In addition, it was found that PVA was able to shield the labile part of VCZ in the complexing medium, which might protect the drug from degradation. To support our assumption, the aqueous solutions containing VCZ/SBEβCD complexes with and without different amounts of PVA were prepared, and the kinetic degradation of VCZ in each sample under a temperature of 40 ± 0.5 °C was evaluated. The temperature-induced degradation of VCZ that followed first-order kinetics based on the linearity was plotted between log (percentage of remaining drug) and time (Table 5).

The observed first-order degradation rate constant (*k_obs_*) was calculated and is presented in Table 5. As expected, VCZ alone was rapidly degraded in aqueous solutions, and the inclusion complex formation with SBEβCD significantly improved the chemical stability of VCZ. Because VCZ was deeply included in the SBEβCD cavity, it remarkedly shielded the drug from degradation. Adding PVA in VCZ/SBEβCD aqueous solutions slightly enhanced the chemical stability of VCZ. The influence of PVA on the decreased degradation rate constant of VCZ was concentration-dependent. It could be explained that PVA reduced the mobility of the VCZ/SBEβCD complex, consequently stabilizing the complex aggregates. On the other hand, the PVA chain occupying the top of the wide rim of SBEβCD, i.e., from MD simulation, may additionally protect the sensitive functional group of VCZ and retard the drug degradation.

### 2.8. Physicochemical and Chemical Characterizations of VCZ Eye Drop Formulations

The aqueous VCZ eye drop formulations were prepared by varying the type and concentrations of PVA. The physicochemical and chemical properties, i.e., pH, viscosity, osmolality and VCZ content, of each formulation are presented in Table 6. The pH of the developed formulations was within a range of 6.70 to 6.98. This pH range is close to the ocular surface pH among humans (7.11 ± 1.5) [44] and, thus, acceptable for topical eye drop formulations. The osmolality of formulations was in a range of 294 to 302 mOsmol/kg, indicating that they were isotonic to the tear fluids [45]. The viscosity of all three formulations was measured at 35 °C and was slightly lower than the experimental values determined at 25 °C [46,47]. As expected, the viscosity of the VCZ formulations depended on the types and concentrations of PVA. The drug content of all formulations was between 100.17 and 100.56%, which indicated that no drug loss occurred during the preparation process. 

### 2.9. In Vitro Mucoadhesion Studies

The mucoadhesive characteristic was investigated by determining the binding efficiency of VCZ eye drop formulations to mucin solution. The percentage of mucoadhesion of tested samples is illustrated in Figure 5A. PVA-H hydrogel had the highest mucoadhesion, followed by PVA-L hydrogel and PVA-L solution, with a statistically significant difference among the samples (*p* < 0.05). Factors affecting mucoadhesion included the functional group, polymer chain flexibility and molecular weight [48]. A large number of hydroxyl groups of PVA can form hydrogen bonds with the hydroxyl or amide groups of mucins. Additionally, increased MW and increased chain length of water-soluble polymers tend to improve mucoadhesion [49,50]. Polymers with a molecular mass larger than 100 kDa have sufficient mucoadhesive properties in biomedical applications due to their favorable entanglement in the mucus layer [51,52]. Our data obtained from VCZ/SBEβCD-loaded PVA-H hydrogel would fit these criteria, thereby resulting in the highest mucoadhesive property. Once this hydrogel adhered to the mucus layer, the adhesion was tight and strong, leading to increased residence time at the ocular surface. This property is required for effective drug permeation through the ocular tissues. Thus, the PVA-H hydrogel was selected for further studies compared with the simple PVA-L solution.

### 2.10. In Vitro Drug Release and Ex Vivo Permeation Studies

Figure 5B demonstrates the cumulative VCZ release profiles of the VCZ/SBEβCD-loaded PVA-based formulations as a function of time. The VCZ release profiles showed a biphasic pattern with an initial rapid release of VCZ (37% from PVA-L solution and 25% from PVA-H hydrogel) within the first 12 h, followed by a sustained manner up to 48 h. Because of the high viscosity and cross-linked density increment from the freeze–thaw cycles, the diffusivity of the PVA-H hydrogel decreased, leading to a slow release rate of VCZ [53]. Both PVA-L solution and PVA-H hydrogel may be suitable for loading VCZ as eye drops, permitting a burst release of the effective drug concentration and a sustained release to reduce the frequency of instillation.

The drug-release kinetic models were calculated and are shown in Appendix A. The release kinetics of both VCZ eye drop formulations were a Fickian diffusion process, which was best fitted by the Higuchi model (R^2^ in a range of 0.932 to 0.947). PVA has a hydrophilic nature, allowing water to permeate easily, leading to swelling of the polymeric matrix. The release profiles of formulations were also fitted with the Korsmeyer-Peppas equation (R^2^ in a range of 0.912 to 0.940). The n values (the diffusional exponent) were in a range of 0.43 < n < 0.85, showing that the anomalous transport pattern was involved and influenced the drug release. Because of their complexity, our developed formulations were governed by more than one mechanism, which could include drug diffusion, relaxation and polymer erosion [54]. These results agreed well with the investigation of Long et al. (2019) [55], in which dexamethasone was released from PVA hydrogel with a similar pattern. 

Figure 5C displays the steady-state flux (*J*) and the apparent permeability coefficient (*P*_app_) values of VCZ eye drop formulations through the porcine cornea. The concentration gradient of drug between the exterior and interior membrane was the rate-limiting step for diffusion of drugs through the membrane. In fact, the porcine cornea comprises lipophilic (epithelium, endothelium) and hydrophilic (stroma) barriers. VCZ must be released from the formulation and be partitioned across these barriers for corneal penetration. Both *J* and *P*_app_ of VCZ from the PVA-L solution were significantly higher than those from PVA-H hydrogel (1.2-fold) (*p* < 0.05). The sustained release manner of VCZ from the cross-linked PVA-H hydrogel network led to decreased VCZ diffusion, thereby lowering the flux through the corneal membrane [56]. 

### 2.11. Irritation Study by Hen’s Egg Test–Chorioallantoic Membrane (HET-CAM)

The HET-CAM test is an alternative method to the Draize test to evaluate ocular irritation [57,58]. The HET-CAM assay is used to predict the toxicity of different excipients or formulation compositions during research and formulation development. The samples’ irritation scores (ISs) were compared with those of C+ (0.1% *w*/*v* sodium hydroxide solution) and C- (0.9% *w*/*v* sodium chloride solution). The test is valid when the negative control does not induce irritation and the positive control causes severe irritation. The IS of C+ was 17.0 ± 0.0, indicating a strong irritant (Figure 5D(a)), while the zero IS value was observed in the case of C- (Figure 5D(b)). The IS value of both formulations was determined to be 0.0. Neither the PVA-L solution nor the PVA-H hydrogel induced hemorrhage, lysis or coagulation (Figure 5D(c,d)), and in that regard, it was similar to C-. It indicated that the developed VCZ eye drop formulations were safe, caused no irritation and were suitable for topical administration of the eye. HET-CAM assay is a useful test for reducing the number of experimental animals. However, for more detailed study of ocular irritation, additional tests, such as corneal cell line viability test and histological examination of excised corneas, should be conducted.

### 2.12. In Vitro Antifungal Activity

The antifungal activities of the intact VCZ, solubilized in dimethyl sulfoxide (DMSO), VFend^®^ and VCZ eye drop formulations against *Aspergillus flavus* (*A. flavus*) and *Fusarium solani* (*F. solani*), were investigated using the microdilution method. Their respective minimum inhibitory concentration (MIC) and minimum fungicidal concentrations (MFC) values were determined and are shown in Table 7. Intact VCZ showed antifungal activity with MIC and MFC values of 0.78 µg/mL in *A. flavus* and 12.5 µg/mL in *F. solani*, respectively. The developed PVA-L solution showed equal MIC and MFC values to intact VCZ solubilized in DMSO. This implies that the ternary VCZ/SBEβCD/PVA complexes improved the aqueous VCZ solubility, which, in turn, caused comparative antifungal activity to the that of the intact VCZ solution. Moreover, it was shown to be superior to that of extemporaneous VFend^®^ eye drops, i.e., binary VCZ/SBEβCD complexes. 

Except for the MFC against *A. Flavus*, which was inferior, the MIC and MFC values of PVA-H hydrogel were comparable with those of the PVA-L solution (two-fold dilution). This observation can be explained by the sustained release of VCZ from the cross-linked PVA network, influencing the fungicidal activity of VCZ. Maw et al. (2022) [59] also reported correlation between controlled release profile and antifungal activity, where a controlled-release Pickering nanoemulsion showed higher MIC values than a conventional nanoemulsion. However, the antifungal activity of PVA-H hydrogel was comparable with that of commercially marketed VFend^®^. As a result, our developed VCZ-loaded PVA hydrogels are potential alternative eye drops that provide fungistatic or fungicidal effects with additional sustained drug release, resulting in decreased instillation frequency and improved patient compliance.

### 2.13. Stability Study

The pH of VCZ eye drop formulations decreased as a function of storage temperature. In the refrigerated condition, the pH of extemporaneous Vfend^®^ eye drops was slightly lower, while the developed VCZ eye drop solutions remained unchanged after storage for 6 months (data not shown). The VCZ contents of eye drop preparations and Vfend^®^ eye drops, initially and after storing for 6 months under various storage conditions, as well as their calculated *k_obs_* and *t*_90_ values, are shown in Table 8. As expected, the *k_obs_* values of all tested samples stored in refrigerated conditions were lower than those of long-term and accelerated conditions. Regarding Vfend^®^ in refrigerated conditions, the *k_obs_* value was 5.62 × 10^−4^ day^−1^ and the *t*_90_ was up to 6 months. However, obvious VCZ degradation was found in long-term conditions and particularly in accelerated conditions, in which the shelf-life was only 44 days. In the case of our developed PVA-L solution, VCZ was stable when stored in a refrigerator and at room temperature but degraded in accelerated conditions. Interestingly, PVA-H hydrogel showed the lowest VCZ degradation and exhibited the maximum *t*_90_ for more than 3 years under a storage temperature at 2 to 8 °C and up to 2.5 years at room temperature. Similarly, Malhotra et al. (2014) [56] studied the stability of a VCZ/HPβCD-complex-loaded PVA solution (1.4% *w*/*v*) and reported a shelf-life of 444 days at room-temperature storage conditions.

According to the data obtained from the stability study, the extemporaneous VFend^®^ containing SBEβCD as a solubilizer to form water-soluble complexes was insufficient to stabilize VCZ in aqueous media. In our case, the addition of PVA maintained the pH of the eye drop solution and preserved the chemical stability of VCZ in an aqueous medium throughout the stability program. Because PVA was able to stabilize the VCZ/SBEβCD complex in an aqueous solution, a greater stability of VCZ eye drops was achieved using PVA with high MW and high concentrations (up to 10% *w*/*v*) (PVA-H hydrogel). Our results show that PVA increases both the solubility and chemical stability of VCZ through the formation of a ternary VCZ/SBEβCD/PVA complex. The result is more effective aqueous VCZ eye drops possessing enhanced shelf-life.

## 3. Materials and Methods

### 3.1. Materials

Voriconazole (VCZ) was kindly donated by Pharma Nueva Co., Ltd. (Bangkok, Thailand). α-Cyclodextrin (αCD), β-cyclodextrin (βCD), γ-cyclodextrin (γCD) and hydroxypropyl β-cyclodextrin (HPβCD) of molecular substitution (MS) 0.65 (mean MW 1400 Da) were kindly contributed by Ashland (Wilmington, DE, USA). Carboxymethyl β-cyclodextrin (CMβCD) MS 0.6 (MW 1312 Da) and randomly methylated β-cyclodextrin (RMβCD) MS 1.8 (MW 1312 Da) were obtained from Wacker Chemie (Burghausen, Germany). Sulfobutyl ether β-cyclodextrin (SBEβCD) MS 0.9 (MW 2163 Da) was kindly donated by Roquette (Lestrem, France). Polyvinyl alcohol (PVA, MW 27,000 Da and 130,000 Da), benzalkonium chloride (BAC) and sodium chloride were obtained from Sigma-Aldrich (St. Louis, MO, USA). Hyaluronic acid (HA) was purchased from Soliance (Pomacle, France). Chitosan hydrochloride salt (CS, Chitoclear, 96% deacetylation degree, MW ≈ 110 kDa) was kindly provided by Primex (Siglufjörður, Iceland). Vfend^®^, sterile powder containing 200 mg of VCZ and 3200 mg of SBEβCD lyophilized complex, was supplied by Pfizer Inc. (New York, NY, USA). Semipermeable cellophane membranes (SpectaPor^®^, molecular-weight cutoff (MWCO) 12–14,000 Da) were bought from Spectrum Europe (Breda, Netherlands). All other chemicals used were of analytical-reagent-grade purity. Water for injection (WFI) was used for the vehicle and Milli-Q (Millipore, Billerica, MA, USA) water was used for the mobile-phase preparation.

### 3.2. Solubility Determinations 

The excess amount of VCZ was added into aqueous CD solutions, i.e., αCD (0–12% *w*/*v*), βCD (0–1.5% *w*/*v*), γCD (0–15% *w*/*v*) and βCD derivatives, i.e., RMβCD, HPβCD, SBEβCD and CMβCD (all 0–10% *w*/*v*). Then, the drug suspensions in sealed vials were heated in a sonicator at 60 °C for 30 min and allowed to cool to room temperature. The suspensions were constantly agitated at 30 ± 1 °C for 7 d. After equilibrium was attained, the resulting samples were filtered through a 0.45 µm nylon filter and the filtrate was analyzed by using high-performance liquid chromatography (HPLC). Each sample was performed in triplicate. The phase-solubility diagrams were constructed by plotting the total dissolved drug concentration (mM) against CD concentration (mM). The K_1:1_ was determined by Equation (1) according to the phase-solubility method [26] and the CE was calculated using Equation (2) [60]:(1)$$K_{1:1}=\frac{\mathrm{slope}}{S_{0}\left(1-\mathrm{slope}\right)}$$
(2)$$\mathrm{CE}=\frac{\mathrm{slope}}{\left(1-\mathrm{slope}\right)}=K_{1:1}\cdot S0$$
where S_0_ is the intrinsic solubility of VCZ.

To investigate the effect of water-soluble polymers on the solubilizing efficiency of VCZ, phase solubility of VCZ in aqueous SBEβCD solutions containing different water-soluble polymers (CS, HA and PVA) was performed. Briefly, CS solution was prepared by dissolving CS in 0.4% *v*/*v* acetic acid solution. For HA or PVA, the polymer was dispersed in purified water and gently heated until completely dissolved. The obtained polymeric solutions were then added to aqueous SBEβCD solutions at concentrations of 1% PVA, 0.1% CS and 0.01% HA (all % *w*/*v*). After that, the excess amount of VCZ was added and the phase solubility was performed via the procedure described above. The K_1:1_ and CE values were then calculated. The CE ratio between the presence and absence of an individual polymer was determined.

### 3.3. Quantitative Determinations

Quantitative determination of VCZ was performed by a reversed-phase HPLC component system from Agilent 1260 Infinity II consisting of liquid chromatography pump (quaternary pump, G7111A), UV-VIS detector (G7115A), auto sampler (G7129A) with Chem Station Software, Version E.02.02 and Shiseido^TM^ Capcell Pack C18 MG II S-5, C18, 150 × 4.5 mm ID with C18 guard cartridge column MGII 5 µm, 4 × 10 mm. The HPLC condition is described below. The mobile phase comprised 50 mM ammonium acetate: acetonitrile (50:50% *v*/*v*); a flow rate of 1.0 mL/min; wavelength of 256 nm; injection volume of 10 μL; oven temperature of ambient temperature; and run time of 7 min.

### 3.4. Particle Size, Size Distribution, Zeta Potential Analysis and Morphological Characterization

VCZ saturated in 10% (*w*/*v*) SBEβCD, with and without PVA (1% *w*/*v*), CS (0.1% *w*/*v*) and HA (0.01% *w*/*v*), was prepared using the heating method (heating in a sonicator at 60 °C for 30 min) and equilibrated at room temperature under constant agitation for 7 days. The sample was further centrifuged (Thermo Fisher Scientific, Waltham, MA, USA) at 4000 rpm at 25 °C for 30 min. The supernatant was collected and subjected to analysis, as described below.

#### 3.4.1. DLS Measurement

The particle size, size distribution and zeta potential of binary VCZ/SBEβCD and ternary VCZ/SBEβCD/polymer (VCZ/SBEβCD/PVA, VCZ/SBEβCD/CS and VCZ/SBEβCD/HA) complexes in aqueous solutions were determined using the DLS technique (Zetasizer^TM^ Nano-ZS Software, Version 7.11, Malvern, UK). The samples were properly diluted with purified water before analysis. The sample was first placed in a cuvette and then in the instrument. Particle size and size distribution and zeta potential were determined at 25 °C and 180° scattering angle. Each sample was performed in triplicate.

#### 3.4.2. TEM Analysis

The morphology of binary VCZ/SBEβCD and ternary VCZ/SBEβCD/PVA complexes was evaluated using TEM (JEOL, JEM-2100F, Peabody, MA, USA). Initially, the sample was placed on a formvar-coated grid. After blotting the grid with filter paper, the grid was transferred onto a drop of negative staining (aqueous 2% phosphotungstic acid solution) and allowed to air dry at room temperature. Finally, the sample was observed under TEM.

### 3.5. Preparation and Characterization of the Binary VCZ/SBEβCD and Ternary VCZ/SBEβCD/PVA Complexes

#### 3.5.1. FT-IR Spectroscopy

Aqueous solutions containing 1:1 molar ratio (m:n; D_m_CD_n_ where m and n represented the total moles of drug and CD, respectively) of the binary complex, i.e., 1:1 molar ratio of VCZ/SBEβCD, with or without PVA (0.1 mM), were prepared by heating in a sonicator at 60 °C for 30 min. The samples were equilibrated at 30 ± 1 °C for 7 days under constant agitation. After equilibrium was attained, the samples were centrifuged (Thermo Fisher Scientific, Waltham, MA, USA) at 4000 rpm for 30 min. Then, the supernatant was withdrawn, frozen at −80 °C for 2 h and lyophilized at −52 °C for 48 h in a freeze-dryer (Labconco Lyophilizer, Kansas City, MO, USA), yielding a solid complex powder (FD). Identical binary PM was prepared by carefully blending ingredients in a mortar with a pestle. The samples were measured in an FT-IR spectrometer (Thermo Scientific Model Nicolet iS10, Waltham, MA, USA) using the attenuated total reflectance technique at room temperature. The samples comprised the following: intact, PM and FD of binary VCZ/SBEβCD complex and FD of the ternary VCZ/SBEβCD/PVA complex. The data were obtained in a range of 400 to 4000 cm^−1^. 

#### 3.5.2. ^1^H-NMR Spectroscopy

The pure solid samples of VCZ, SBEβCD, PVA as well as 1:1 molar ratio of binary VCZ/SBEβCD and ternary VCZ/SBEβCD/PVA complexes were dissolved in dimethyl sulfoxide-*d*_6_ (DMSO-*d*_6_). To form CD inclusion complexes, the samples were equilibrated at 30 ± 1 °C for 24 h before being subjected to analysis. ^1^H-NMR spectroscopy measurement was performed using a 500 MHz ^1^H-NMR spectrometer (BRUKER Model AVANCE III HD, Billerica, MA, USA). The spectra and chemical shift values (*δ*) were recorded as ppm. The residual solvent signal (2.5000 ppm) was used as the internal reference and the chemical shift values (Δ*δ**) were calculated according to Equation (3):(3)$$\Delta \delta^{*}=\delta_{\left(complex\right)}-\delta_{\left(free\right)}$$

### 3.6. MD Simulations

The starting structure of VCZ (SID: 388929365) was downloaded from PubChem database [61], where the monomer of PVA was generated by the Gaussian09 Program [62]. After that, both structures were optimized using the HF/6-31G* level of theory and Gaussian09 Program. The polymer of PVA, including 50 units of PVA monomers, was built using the LEaP module implemented in the AMBER16 package [63]. The SBEβCD with degree of substitution (DS) of 7 at the primary rim was obtained from our previous work [64]. 

The VCZ/SBEβCD complex did not display crystal structure in the crystallographic data and, thus, the docking procedure was applied to obtain a starting point for the calculations. The inclusion complexes of VCZ/SBEβCD and VCZ/SBEβCD/PVA were generated by docking protocol using CDOCKER module in the Accelrys Discovery Studio 2.5 (Accelrys Software Inc., San Diego, CA, USA). From molecular docking, two main possible docked inclusion complexes, difluoro phenyl insertion (Form I) and fluoropyrimidine insertion (Form II) (near the secondary rim of SBEβCD), of each system were used as the initial inclusion complexes for MD simulations, as depicted in Appendix A. 

MD simulations of all models were carried out using the AMBER16 Program Package [63]. The partial atomic charges and parameters of VCZ and PVA as well as sulfobutyl ether group were constructed using the previous standard procedures [65,66,67]. The general AMBER force field [68] was applied for the VCZ, PVA and sulfobutyl ether groups, whereas the βCD moiety was treated with the Glycam06j carbohydrate force field [69]. The TIP3P water model [70] was used to solvate each inclusion complex, with a spacing distance of 10 Å. Seven Na^+^ ions were then added to neutralize the system. Afterward, the added water molecules were minimized using 1000 steps of steepest descent (SD) and continued by 3000 steps of conjugated gradient (CG). Then, all molecules were minimized using the same procedures. All models were heated from 10 K to 298 K with a constant volume ensemble (NVT) for 100 ps and followed by all-atom MD simulations with a constant pressure ensemble (NPT) at 1 atm and 298 K for 100 ns. The SHAKE algorithm [71] was used to constrain all chemical bonds involving hydrogen. The particle mesh Ewald’s method [72] was employed to treat the long-range electrostatic interactions with a 12 Å cutoff. The equilibrium status was determined using root mean squared displacement (RMSD). The size of the inclusion complex and the solvation effect of all studied models were investigated using radius of gyration (R_gyr_) and solvent-accessible surface area (SASA) calculations, respectively. Furthermore, the ligand-binding affinities of the VCZ/SBEβCD and VCZ/SBEβCD/PVA inclusion complexes were estimated using the molecular mechanics/generalized Born surface area (MM/GBSA) and the molecular mechanics/Poisson–Boltzmann surface area (MM/PBSA) binding free energy (Δ*G_bind_*) calculations [73]. 

### 3.7. Kinetic Degradation Studies 

To evaluate the chemical stability of VCZ in aqueous solutions, the kinetic degradation of VCZ in pure water and in aqueous 10% *w*/*v* SBEβCD solutions in the absence and presence of various concentrations of PVA (1, 2 and 5% *w*/*v*) was performed. The amount of VCZ solubilized in each medium was estimated from the phase-solubility diagrams. The samples were stored at 40 ± 0.5 °C and withdrawn at 0, 24, 36, 48, 60, 72, 84, 96 and 168 h intervals. The collected samples were determined regarding VCZ content by HPLC. The logarithm of percent VCZ remaining was plotted against time and kinetic parameters, i.e., the observed first-order reaction rate constant (*k_obs_*) was calculated from the equation according to the respective order of linear regression of the correlation coefficient (R).

### 3.8. Preparation of VCZ Eye Drop Formulations

The VCZ eye drop formulations, i.e., solutions and hydrogels, were developed based on the type and concentrations of PVA. Firstly, the aqueous PVA solutions were prepared by dispersing PVA ((PVA-L) MW 27,000 Da or (PVA-H) MW 130,000 Da) in WFI containing 0.02% (*w*/*v*) BAC. The resulting solutions were then heated at 90 °C until completely dissolved and cooled to room temperature. VCZ (0.5 gm) and SBEβCD (6.0 gm) were dissolved in an aqueous PVA medium and stirred until a homogenous solution was obtained. After that, the desired volume was filled up with the same medium and, finally, the preparation was adjusted to isotonicity with sodium chloride. In the case of VCZ/SBEβCD-loaded PVA hydrogels, the preparation process was further performed by the freeze–thaw method [74]. Two consecutive freeze–thaw cycles were completed; each cycle consisted of 12 h of freezing at −20 °C followed by 12 h of thawing at 25 °C. The compositions of VCZ eye drop formulations are shown in Table 9. The VCZ eye drop formulations were designated as PVA-L solution, PVA-L hydrogel and PVA-H hydrogel according to the types and concentrations of PVA. 

### 3.9. Characterizations of VCZ Eye Drop Formulations

#### 3.9.1. pH, Viscosity and Osmolality

The pH of formulations was measured using a pH meter (Mettler Toledo™, Seven Compact, Gießen, Germany) at 25 °C. The osmolality of formulations was measured by an osmometer (Genotec, OSMOMAT 3000 basic, Berlin, Germany) at room temperature by using the freezing point depression principle. The instrument was calibrated with standard sodium chloride solutions before analysis. The viscosity determination of each formulation was conducted using a viscometer (A&D Company, Sine-wave Vibro SV-10, Tokyo, Japan) at 25 and 35 °C. All measurements were performed in triplicate.

#### 3.9.2. Determination of Drug Content 

Total VCZ content in eye drop formulations was determined by diluting 100 µL of sample with a mixture of mobile phase and then filtered. The amount of VCZ in the filtrate was analyzed by HPLC. All measurements were performed in triplicate.

### 3.10. In Vitro Mucoadhesion Study

The mucoadhesive characteristics of VCZ eye drop formulations were determined by a modified method [75]. Briefly, the 0.2% (*w*/*v*) aqueous mucin solution (porcine stomach, Type II) was prepared in simulated tear fluid (STF), pH 7.4 (composition (100 mL): NaCl 0.68 g, NaHCO_3_ 0.22 g, CaCl_2_⋅2H_2_O 0.008 g, KCl 0.14 g). The test samples were mixed with 1 mL of mucin solution and incubated at 35 °C for 30 min and kept for 24 h at room temperature. Then, the samples were ultracentrifuged at 18,000 rpm at 4 °C for 1 h. After that, the supernatant was collected and free mucin was quantified using a UV spectrophotometer at 251 nm. Before the study, all tested samples were scanned at 251 nm and no interference at mucin UV wavelength was observed. The binding efficiency, i.e., % mucoadhesion, of mucin with samples was calculated using Equation (4).
(4)$$\%\;Mucoadehsion=\frac{C_{total}-C_{supernatant}}{C_{total}}\times 100$$
where *C_total_* is total mucin concentration and *C_supernatant_* is mucin concentration in supernatant.

### 3.11. In Vitro Drug-Release Study

The in vitro release of VCZ from eye drop preparations through a semipermeable membrane (MWCO 12–14,000 Da) was investigated using Franz diffusion cell apparatus. The receptor phase consisted of phosphate-buffer saline, pH 7.4, containing 2.5% (*w*/*v*) SBEβCD. SBEβCD was added to the receptor medium to maintain a sink condition throughout the study. The receptor medium was sonicated to remove the dissolved air before the study. The sample (1.5 mL) was placed in the donor phase, the receptor medium (12 mL) was continuously stirred at 150 rpm and the temperature was maintained at 35 ± 0.5 °C. A 150 μL aliquot of receptor medium was withdrawn at time intervals of 0.5, 1, 2, 3, 4, 5, 6, 8, 12, 24 and 48 h and replaced by an equal volume of fresh receptor medium. The VCZ content was analyzed by HPLC. The amount of VCZ cumulative release was plotted against time. Each sample was performed in triplicate. For further information, the drug-release kinetics were investigated by fitting to various models, i.e., zero-order, first-order, Higuchi, Hixson-Crowell and Korsmeyer–Peppas models [76].

### 3.12. Ex Vivo Permeation Study

The ex vivo method was used to evaluate and predict VCZ permeation from the eye drop formulations in the anterior segment of the eye after topical application. The permeation studies were conducted across the cornea isolated from porcine eyes obtained from the Faculty of Veterinary Science, Chulalongkorn University. In this study, the cornea that was dissected from the eyes, which was obtained within 2 h after the death of the animals, replaced the semipermeable membrane in the previously described in vitro drug-release studies. VCZ permeation from each formulation was determined in quadruplicate and the reported values were the mean values ± standard deviation (S.D.). The *J* and the *P*_app_ were calculated from Equation (5):(5)$$J=\frac{dq}{A\cdot dt}=P_{\mathrm{app}}\cdot C_{d}$$
where *A* is the surface area of the mounted membrane 1.77 cm^2^ and *C_d_* is the initial concentration of the drug in the donor chamber. The *J* was calculated as the slope of linear plots of the amount of drug in the receptor chamber (*q*) versus time (*t*).

### 3.13. HET-CAM Test

To study the cytocompatibility of VCZ eye drop formulations, the HET-CAM assay was carried out according to the Interagency Coordinating Committee on the Validation of Alternative Methods (ICCVAM)—Recommended Test Method Protocol: HET-CAM Test Method, using fertile broiler chicken eggs [77]. Firstly, eggs were hatched for 9 days in an automatic rotation incubator at 38.0 ± 0.5 °C and 58.0 ± 2.0% relative humidity (RH). During the last 24 h of the incubation period, the rotation was stopped to locate the air sac in the wider part of the egg. The assay was performed on the 9th incubation day. The outer shell of the eggs was opened. After that, 300 µL of VCZ eye drop formulations, i.e., PVA-L solution or PVA-H hydrogel, as well as positive control (C+), i.e., 0.1% (*w*/*v*) sodium hydroxide solution or negative control (C-), i.e., 0.9% (*w*/*v*) sodium chloride solution were directly applied onto the chorioallantoic membrane (CAM) surface. The CAM was observed over a period of 0.5, 2 and 5 min. The irritation scores were marked from 0 to 21 according to Luepke (1985) [78] and irritation was classified as follows: (I) hemorrhage (bleeding from the vessels), (II) vascular lysis (blood vessel disintegration) and (III) coagulation (intra- and/or extravascular protein denaturation). The experiments were performed in triplicate. 

### 3.14. In Vitro Antifungal Activity

Regarding the antifungal activity of VCZ eye drops, 0.5% (*w*/*v*) Vfend^®^ eye drops (reconstituted Vfend^®^ powder: commercially available for intravenous administration) and VCZ solubilized in DMSO against *A. flavus* (DMST 70785) and *F. solani* (DMST 70786) were determined using broth microdilution assay according to the protocol of the Clinical and Laboratory Standards Institute (CLSI) [79]. Briefly, the fungi were subcultured onto Sabouraud dextrose agar (SDA) slants and the resulting organism suspensions were stored at 2 to 8 °C. Then, the organism suspensions were further diluted with 0.85% saline to obtain a cell density of 1 × 10^6^ to 5 × 10^6^ cells/mL. Each well was inoculated with 50 µL of two-fold dilution inoculum suspension. An aliquot of 50 µL of test samples was placed in separate wells. DMSO and the drug-free medium were included as growth controls. All the samples were performed in triplicate. The plates were incubated at 35 °C for 96 to 120 h. MICs were read and defined as the lowest sample concentration at which no growth could be observed. After MIC readings, 10 µL aliquots were removed from each growth-negative well and spread on SDA petri dishes. The plates were incubated at 35 °C. After 4 to 7 days of the incubation period, the fungal colonies were counted. The MFCs were defined as the lowest sample concentration from which no colony was visible on the agar plate.

### 3.15. Stability Study

The stability of developed VCZ eye drop formulations was evaluated by following the International Conference on Harmonization (ICH) guidelines [80], with modifications. For comparison, 0.5% (*w*/*v*) Vfend^®^-eye drops were also determined. All VCZ eye drop formulations were stored in well-stoppered glass containers at 4 °C, long-term (30 ± 2 °C at 60 ± 5% RH) and accelerated (40 ± 2 °C at 75 ± 5% RH) conditions. At time intervals of 0, 7, 14, 21, 30, 90 and 180 days, the samples were taken, and pH and drug content were determined. All measurements were performed in triplicate. 

### 3.16. Statistical Analysis

All quantitative data were presented as mean ± S.D. One-way ANOVA with Tukey post hoc test was determined using SPSS, Version 16.0. A *p*-value *p* < 0.05 was considered as statistical significance.

## 4. Conclusions

Among the CDs tested, SBEβCD was the most effective CD solubilizer of VCZ-forming water-soluble VCZ/SBEβCD complexes. Adding PVA to form VCZ/SBEβCD/PVA ternary complex aggregates was found to be effective regarding solubilization and stabilization of VCZ/SBEβCD complexes. The developed VCZ/SBEβCD-loaded PVA eye drop formulations exhibited good physicochemical properties and high mucoadhesion. The drug-release profiles showed a burst release followed by a sustained drug delivery, and the released VCZ was able to permeate through the porcine cornea. These formulations showed no signs of irritation and demonstrated promising antifungal activity against *A. flavus* and *F. solani*, comparable with those of the extemporaneous Vfend^®^ eye drops. The stability studies of VCZ eye drop formulations showed that the formulation comprised high MW and high concentrations of PVA, i.e., PVA-H hydrogel, preserving the chemical stability of VCZ in aqueous-based formulation, with a long tentative shelf-life. When compared with the extemporaneous Vfend^®^ eye drops, the VCZ/SBEβCD-loaded PVA hydrogel was found to be successful in improving chemical stability. Furthermore, the addition of PVA in our developed VCZ formulation required less SBEβCD (6% *w*/*v*) than comparable eye drops, which contained 8% *w*/*v* and, thus, it may be advantageous in terms of toxicology and manufacturing costs. In vivo testing of the eye drops will be conducted in further studies.

## Figures and Tables

**Figure 1 ijms-24-02343-f001:**
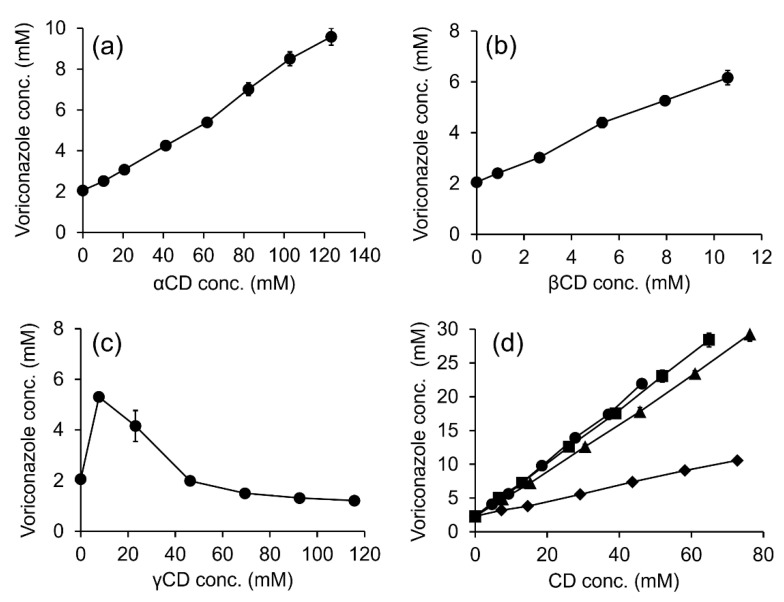
Phase-solubility profiles of VCZ in aqueous CD solutions; αCD (**a**), βCD (**b**), γCD (**c**) and βCD derivatives, i.e., SBEβCD (●), HPβCD (■), RMβCD (▲) and CMβCD (♦) (**d**).

**Figure 2 ijms-24-02343-f002:**
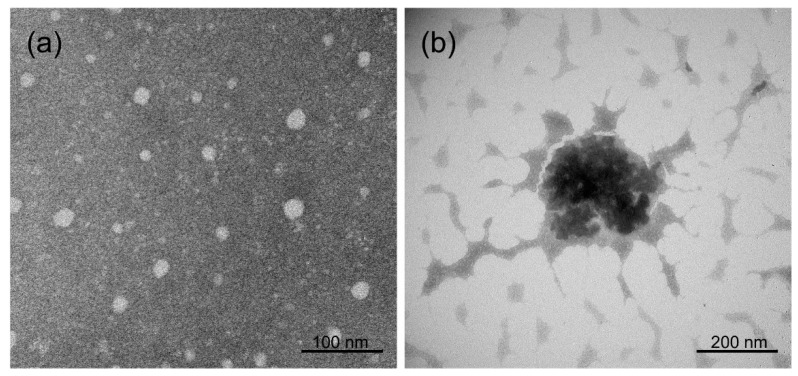
TEM images of aqueous 10% (*w*/*v*) SBEβCD solutions saturated with VCZ with and without PVA (**a**) VCZ/SBEβCD and (**b**) VCZ/SBEβCD/PVA.

**Figure 3 ijms-24-02343-f003:**
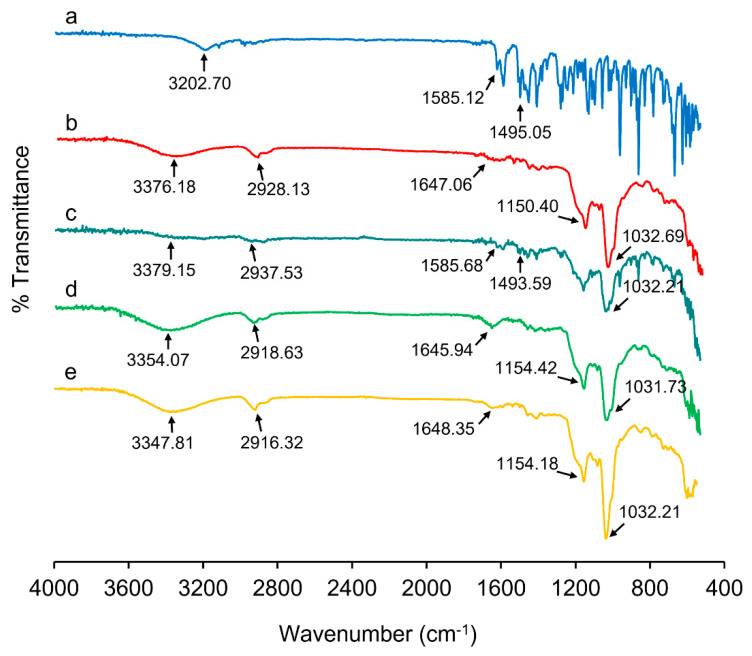
FT-IR spectra of (**a**) VCZ, (**b**) SBEβCD, (**c**) PM VCZ/SBEβCD, (**d**) FD VCZ/SBEβCD and (**e**) FD VCZ/SBEβCD/PVA.

**Figure 4 ijms-24-02343-f004:**
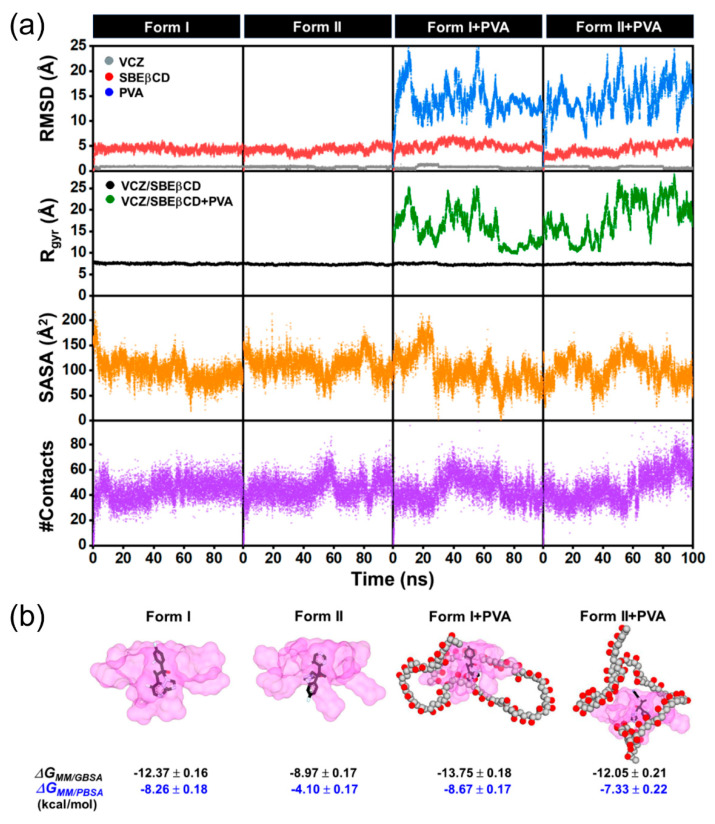
(**a**) RMSD, R_gyr_, SASA and number of atom contacts of two different forms of the VCZ/SBEβCD and VCZ/SBEβCD/PVA inclusion complexes along the simulation time, (**b**) representative snapshots of the binding orientation of each system with the average *ΔG_MM/GBSA_* and *ΔG_MM/PBSA_* in kcal/mol calculated from over the last 20 nanoseconds (ns) simulation time.

**Figure 5 ijms-24-02343-f005:**
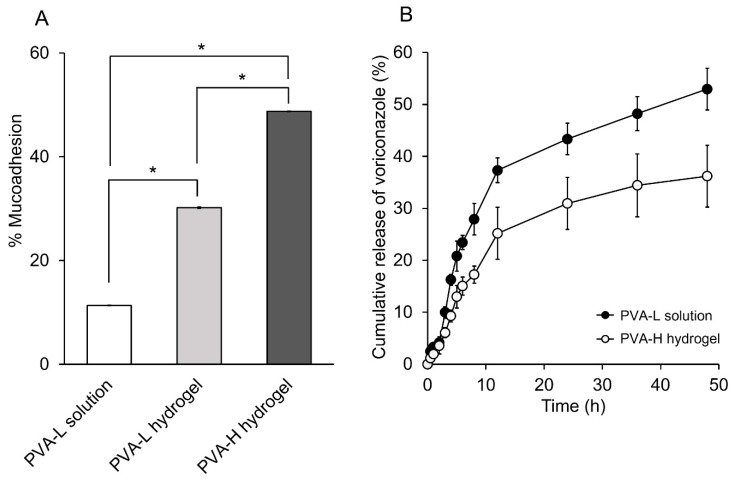
(**A**) Percentage of mucoadhesion of PVA-L solution, PVA-L hydrogel and PVA-H hydrogel (mean ± S.D., n = 3) (* *p* < 0.05); (**B**) in vitro release study of VCZ from PVA-L solution and PVA-H hydrogel (mean ± S.D., n = 3); (**C**) permeation flux (*J*) and apparent permeability coefficient (*P*_app_) of VCZ-loaded PVA-L solution and PVA-H hydrogel through the porcine cornea (mean ± S.D., n = 4) (* *p* < 0.05); (**D**) images of HET-CAM at 5 min post-instillation for the different samples; (**a**) sodium hydroxide (C+); (**b**) sodium chloride (C-); (**c**) PVA-L solution; and (**d**) PVA-H hydrogel.

**Table 1 ijms-24-02343-t001:** Chemical structure and some physicochemical properties of voriconazole.

Physicochemical Properties	Voriconazole
Chemical structure	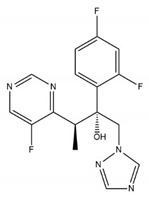
Molecular Weight (g/mol)	349.3
Melting point (°C) ^a^	132.4
pKa	1.76
logP_o/w_ ^b^	1.65
S_0_ in water (at RT ^c^)	0.5 mg/mL

^a^ dec. = decomposition upon heating; ^b^ logarithm of the octanol/water partition coefficient; ^c^ RT = room temperature (at 25 °C).

**Table 2 ijms-24-02343-t002:** Apparent stability constants (K_1:1_) and the complexation efficiency (CE) values of VCZ/CD complexes in aqueous solutions with and without individual polymers at 30 °C ± 1 °C (mean, n = 3).

Cyclodextrin	Polymers	Type	R^2^	K_1:1_ (M^−1^)	CE	CE ratio ^b^
αCD	-	A_L_	0.9947	30.3	0.066	-
βCD	-	A_L_	0.9946	301.8	0.658	-
γCD	-	B_S_	- ^a^	- ^a^	- ^a^	-
RMβCD	-	A_L_	0.9993	250.1	0.545	-
HPβCD	-	A_L_	0.9996	306.6	0.668	-
CMβCD	-	A_L_	0.9989	59.6	0.130	-
SBEβCD	-	A_L_	0.9987	338.1	0.737	1.00
SBEβCD	1% PVA	A_L_	0.9957	412.4	0.950	1.30
SBEβCD	0.01% HA	A_L_	0.9987	339.3	0.781	1.06
SBEβCD	0.1% CS	A_L_	0.9918	355.1	0.818	1.12

^a^ could not be calculated; ^b^ CE of the drug in the presence of water-soluble polymer/CE of the drug when no polymer was present.

**Table 3 ijms-24-02343-t003:** Mean particle size, size distribution and zeta potential of binary VCZ/SBEβCD and ternary VCZ/SBEβCD/polymer complex aggregates (mean ± S.D., n = 3).

Cyclodextrin (%w/v)	Polymer (%w/v)	Mean Particle Size (nm)	Intensity (%)	PDI ^a^	Zeta Potential (mV)
10% SBEβCD	-	1.34 ± 0.43	58.30 ± 33.44	0.21 ± 0.07	−1.01 ± 2.35
		11.24 ± 8.55	23.16 ± 18.48		
		719.92 ± 315.74	18.53 ± 12.98		
	0.1% CS	22.70 ± 6.87	9.98 ± 4.63	0.53 ± 0.02	+2.75 ± 1.33
		240.73 ± 81.13	89.98 ± 7.04		
		4269 ± 656.91	2.87 ± 1.59		
	0.01% HA	1.17 ± 0.10	48.74 ± 22.01	0.68 ± 0.35	−1.73 ± 4.70
		977.22 ± 321.61	46.10 ± 24.08		
		1546.17 ± 222.57	7.71 ± 4.54		
	1% PVA	1.14 ± 0.10	14.33 ± 1.34	0.60 ± 0.06	−3.15 ± 2.46
		23.55 ± 2.56	63.20 ± 5.10		
		366.20 ± 97.59	20.44 ± 4.72		

^a^ PDI = polydispersity index.

**Table 4 ijms-24-02343-t004:** The ^1^H chemical shifts of VCZ and SBEβCD in binary VCZ/SBEβCD and ternary VCZ/SBEβCD/PVA complexes.

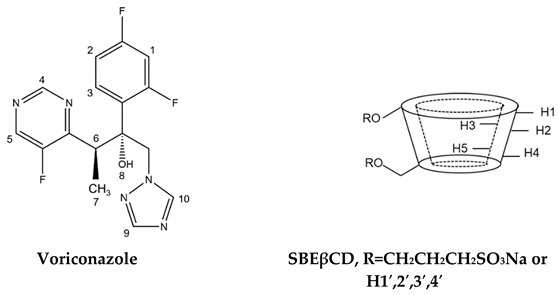		
^1^H Assignment	Δδ* = (δ_complex_ − δ_free_)	
	VCZ/SBEβCD	VCZ/SBEβCD/PVA
*VCZ*		
H-1	−0.006	−0.006
H-2	−0.004	−0.004
H-3	−0.004	−0.001
H-4	−0.005	−0.005
H-5	−0.005	−0.005
H-6	−0.005	−0.011
H-7	−0.002	−0.002
H-9	−0.003	−0.003
H-10	+0.003	+0.003
*SBEβCD*		
H-1	+0.001	+0.001
H-2,4	+0.002	−0.005
H-3	+0.002	+0.009
H-5	+0.004	+0.022
H-6	- ^a^	- ^a^
H-2′,H-3′	- ^a^	+0.001

^a^ Could not calculate due to overlapping of signal; Δδ* = chemical shift values.

**Table 5 ijms-24-02343-t005:** Degradation rate constants (*k_obs_*) of VCZ in pure water and aqueous SBEβCD solutions in the absence and presence of PVA (1, 2 or 5% w/v) under storage temperature at 40 ± 0.5 °C (mean, n = 3).

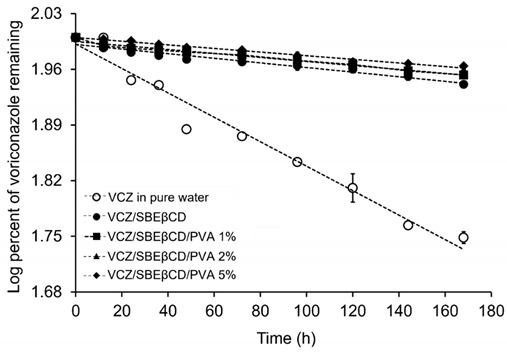			
Drug	Cyclodextrin	Polymer	*k_obs_* (h^−1^) at 40 °C
Voriconazole	-	-	3.53 × 10^−3^
	SBEβCD	-	6.70 × 10^−4^
		1% PVA	5.79 × 10^−4^
		2% PVA	5.55 × 10^−4^
		5% PVA	5.27 × 10^−4^

**Table 6 ijms-24-02343-t006:** The pH value, osmolality, viscosity and drug content of VCZ eyedrop formulations (mean ± S.D., n = 3).

Parameter		Formulation	
	PVA-L Solution	PVA-L Hydrogel	PVA-H Hydrogel
pH	6.98 ± 0.35	6.71 ± 0.30	6.70 ± 0.07
Osmolality (mOsmol/kg)	302.3 ± 3.5	306.7 ± 10.0	294.7 ± 2.51
Viscosity at 25 °C (cPs)	8.46 ± 0.78	26.73 ± 1.05 ^a^	160.00 ± 13.00 ^a^
Viscosity at 35 °C (cPs)	5.36 ± 0.12	19.56 ± 2.38 ^a^	109.13 ± 25.30 ^a^
Drug content (%)	100.17 ± 1.46	100.47 ± 2.13	100.57 ± 3.17

^a^ Significant difference when compared with PVA-L solution (*p* < 0.05); cPs = centipoise.

**Table 7 ijms-24-02343-t007:** In vitro antifungal activities of VCZ in DMSO, Vfend^®^ eye drops and VCZ/SBEβCD-loaded PVA-L solution and PVA-H hydrogel against fungal species (mean, n = 3).

Sample	*A. flavus*		*F. solani*	
	MIC ^a^	MFC ^b^	MIC ^a^	MFC ^b^
VCZ solubilized in DMSO	0.78	0.78	12.5	12.5
Vfend^®^-eye drops	0.78	3.10	25.0	25.0
PVA-L solution	0.39	0.70	12.5	12.5
PVA-H hydrogel	0.39	3.10	25.0	25.0

^a^ MIC = minimal inhibitory concentration (µg/mL); ^b^ MFC = minimal fungicidal concentration (µg/mL).

**Table 8 ijms-24-02343-t008:** Drug content, degradation rate constants and shelf-lives of Vfend^®^ eye drops and VCZ eye drop formulations stored for up to 6 months.

Formulations	VCZ Content		Degradation Rate Constant (*k_obs_*, Day^−1^)	*t*_90_ (Days)
	Initial	6 Months		
**2–8 °C**				
Vfend^®^ eye drops	102.15 ± 5.08	92.49 ± 5.08	5.62 × 10^−4^	183
PVA-L solution	100.17 ± 1.46	94.62 ± 1.44	3.48 × 10^−4^	301
PVA-H hydrogel	100.56 ± 3.17	98.65 ± 1.41	8.95 × 10^−5^	1172
**Long term (30 ± 2 °C, 60 ± 5% RH ^a^)**				
Vfend^®^ eye drops	102.15 ± 5.08	86.39 ± 2.43	1.00 × 10^−3^	105
PVA-L solution	100.17 ± 1.46	93.72 ± 1.13	3.90 × 10^−4^	267
PVA-H hydrogel	100.56 ± 3.17	98.36 ± 0.59	1.03 × 10^−4^	1015
**Accelerated (40 ± 2 °C, 75 ± 5% RH ^a^)**				
Vfend^®^ eye drops	102.15 ± 5.08	68.27 ± 0.86	2.40 × 10^−3^	44
PVA-L solution	100.17 ± 1.46	87.53 ± 3.11	8.60 × 10^−4^	123
PVA-H hydrogel	100.56 ± 3.17	98.18 ± 0.91	1.16 × 10^−4^	903

^a^ RH = relative humidity.

**Table 9 ijms-24-02343-t009:** Compositions of VCZ eye drop formulations.

Formulations	Compositions (% *w*/*v*) ^a^			
	VCZ	SBEβCD	PVA (LMW) ^b^	PVA (HMW) ^c^
PVA-L solution	0.5	6	5	-
PVA-L hydrogel	0.5	6	10	-
PVA-H hydrogel	0.5	6	-	10

^a^ Each formulation, all % *w*/*v* and the formulations were adjusted with water for injection to 100 mL. All formulations were adjusted with sodium chloride to obtain isotonicity; ^b^ PVA (LMW): polyvinyl alcohol (low molecular weight 27,000 Da); ^c^ PVA (HMW) polyvinyl alcohol (high molecular weight 130,000 Da).

## Data Availability

Not applicable.

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
