# Peer review of "Voriconazole Eye Drops: Enhanced Solubility and Stability through Ternary Voriconazole/Sulfobutyl Ether β-Cyclodextrin/Polyvinyl Alcohol Complexes"

_ijms, 2023, doi:10.3390/ijms24032343_

Round 1

Reviewer 1 Report

Dear Authors,

The article 'Voriconazole eye drops: enhanced solubility and stability through ternary voriconazole/sulfobutyl ether β-cyclodextrin/polyvinyl alcohol complexes' is a remarkable study suitable for the special issue of the journal.

The authors have prepared the abstract and introduction part of the article in a satisfactory and explanatory manner. Similarly, the materials and methods used are explained in detail.

However, I have indicated in detail some of the points I have identified for the article, which should be revised.

- Some simple typos attract attention in the tables/figures. It would be appropriate to review and edit them. (For example, Table 1 has punctuation errors in its content, Table 2 has shifting columns, and Figure 1 legends have punctuation errors).

-It would be helpful to clearly state the abbreviations used in all tables and figures as footnotes. (e.g. table 8)

- In the writing guide of the journal, in a way that does not comply with the rules specified in the references section, while some literature information is given, the name of the year is specified in parentheses after the referenced author (For example, on line 369 ….Long et al. (2019)……). It should be appropriate to revise and edit it.

- Findings and discussion are given under the same title in the publication. I was still determining whether this situation was suitable for the writing rules of the journal. The journal's writing guide states that the 'Results and discussion' sections should be under a separate heading. 

- Under the heading of findings and discussion, '3.8. Physicochemical and chemical characterizations of VCZ eye drop formulations' seem to have a numbering style problem with this title (Line 308…).

- In general, the findings are presented in detail under the heading of findings and discussion. However, it created the impression that more discussion of the results needed to be presented. It is necessary to discuss the findings in more detail. For example;

2.9. Only data are presented under the HAt-CAM test title; no information for discussion is observed.

o 2.10. in vitro antifungal activities….only the findings are given. However, a discussion was not written with the results obtained according to the literature.

o Similarly, only data are presented under the title of '2.11.- stability studies…, no information for discussion is observed.

 Kind regards

Reviewer 2 Report

The article presents a thorough analysis of the problem in question and delivers sound results. The work encompasses a good diversity of methods including much more than just a structural analysis of a new formulation. 

I have only one remark regarding the computational part. Here, a standard method has been applied: docking with CDDOCKER - energy optimization - MD - MMGBSA. Indeed, for a long time this has been a standard way of approaching such systems.

However, this approach is not a perfect one and often leads to false results. The reason is the first step. Application of the CDDOCKER or AutoDock in most of cases is questionably accurate. There are two ways of dealing with such an issue. The first one is to use a high level of quantum mechanic (QM) energy optimization, meaning DFT with a bigger basis set than 6-31G* [for instance B3LYP/6-31G(d,p) ] and an inclusion of a solvent (e.g. PCM solvent model). In this work, I would expect that Authors obtained rather low level of accuracy by using HF/6-31G*. 

Another and the perfect option, is to use a crystal structure of the complex as a starting point. The solid structure may differ from the one in a solution, however 1) it can be checked experimentally 2) it always has a higher probablity of being a real one when compared to a relatively random guess of CDDOCKER.

From what I have checked, as of now, in the crystallographic data base CCDC, there is no voriconazole-SBE-beta-CD structure. However, having done such an effort by constructing this research composed of a great variety of methods, next time, I would suggest doing a bit of an additional effort and try to obtain a crystal of the complex, do the SXRD analysis and in cooperation with a crystallographer obtain the crystal structure. Or simply ommit the molecular modelling part, as starting from a possibly wrong structure does not lead anywhere. It is true that the QM calculations can strongly influence the docked system, however they must be performed at the sufficient level of theory and the solvent presence should be considered. 

All in all, I find this work to be a thorough analysis, well-constructed and well-discussed with a real advantages to the pharmaceutical field. 

The standard molecular medthod has been applied properly but nowadays we already know that the standards must often undergo a reconstruction. In this particular case, I would suggest to add (in ''methods'' section) an information that due to a lack of a crystal structure of a complex deposited in the crsystallography data base which could have served as a starting point for the calculations, the docking procedure has been applied. This will change nothing in this particluar work but will be an important piece of data for the readers of the paper who will consider performing a similar type of experiments for their systems. The standard molecular modelling approaches must be revisited and there is no better way to do that than by including such information in the papers published today.
